# Reinforcing Effect of Polypropylene Waste Strips on Compacted Lateritic Soils

**Régis Marçal** [1]**, Paulo César Lodi** [1,]*****, Natália de Souza Correia** [2]**, Heraldo Luiz Giacheti** [1]**, Roger Augusto Rodrigues** [1] **and John S. McCartney** [3]

[1] Department of Civil and Environmental Engineering, São Paulo State University (UNESP),
   Av. Engenheiro Luiz Edmundo Carrijo Coube 14-01, Bauru, SP 17033-360, Brazil;
   regis-ata@hotmail.com (R.M.); h.giacheti@unesp.br (H.L.G.); roger.rodrigues@unesp.br (R.A.R.)

[2] Department of Civil Engineering, Federal University of Sao Carlos (UFSCar), Rodovia Washington Luiz,
   São Carlos, SP 17033-360, Brazil; ncorreia@ufscar.br

[3] Department of Structural Engineering, University of California at San Diego (UCSD), 9500 Gilman Dr.,
   SME 442J, La Jolla, CA 92093-0085, USA; mccartney@ucsd.edu

***** Correspondence: paulo.lodi@unesp.br; Tel.: +55-646-755-2239

**Abstract:** This study evaluated the strength properties of compacted lateritic soils reinforced with polypropylene (PP) waste strips cut from recycled plastic packing with the goal of promoting sustainability through using local materials for engineering work and reusing waste materials as low-cost reinforcements. Waste PP strips with widths of 15 mm and different lengths were uniformly mixed with clayey sand (SC) and clay (CL) soils with the goal of using these materials as low-cost fiber reinforcements. The impact of different PP strip contents (0.25% to 2.0%) and lengths (10, 15, 20, and 30 mm) on the unconfined compressive strength (UCS) of the soils revealed an optimum combination of PP strip content and length. Statistical analysis showed that PP strip content has a greater effect than the PP strip length on the UCS for both soils. Results led to the definition of an empirical equation to estimate the UCS of strip-reinforced soils. The results from direct shear tests indicate that the SC soil showed an increase in both apparent cohesion and friction angle after reinforcement, while the CL soil only showed an increase in friction angle after reinforcement. California bearing ratio (CBR) tests indicate that the SC soil experienced a 70% increase in CBR after reinforcement, while the CBR of the CL soil was not affected by strip inclusion.

**Keywords:** soil improvement; polypropylene strips; geotechnical properties; sustainable reuse of plastic waste

## 1. Introduction

Finding new ways to recycle plastic waste from water bottles, disposable cups, plates, or plastic packaging for foods has become a major challenge worldwide. According to the World Economic Forum (2016), a million plastic bottles are bought around the world every minute, and this number may jump 20% by 2021, potentially leading to an environmental disaster. As also pointed out in this report, plastic production has increased from 15 million tons in the 1960s to 311 million tons in 2014 and is expected to triple by 2050. Furthermore, the 2030 Agenda for Sustainable Development [1] sets out in its goals a substantial reduction in waste generation through recycling, reduction, and reuse and encourages the use of local materials in engineering works.

Environmental challenges have stimulated researchers to find techniques to improve the strength properties of geotechnical materials [2]. In the context of alternative or recycled waste materials in soil improvement, tire shreds and rubber fibers have been extensively studied [3–10]. Further, the use

of fiber reinforcement, especially with local soils, has been recognized as a viable technique for soil improvement in numerous geotechnical engineering applications. Fiber reinforcement has been used in a range of applications, including as backfill in retaining structures, stabilization of subgrade and subbases, improvement in soil bearing capacity, reinforcement of soft soil embankments, control of soil hydraulic conductivity, improvement of erosion resistance, piping prevention, and shrinkage crack mitigation [11–15]. Fiber reinforcements can carry tensile stresses, which are mobilized by friction between the reinforcements and the soil. The mobilization of tensile stresses in the reinforcements generally leads to an increase in the shear strength of the soils, namely that generated by redistributing shear stresses in soils through their tensile strength. Randomly distributed polymeric additions, such as polypropylene (PP) and polyethylene terephthalate (PET), incorporated in soils improve their mechanical behavior.

Gathering the idea of plastic recycling and soil improvement, Consoli et al. [16] carried out one of the first experiments on the utilization of the polyethylene (PET) fibers derived from plastic wastes (stretched cylindrical shapes) in the reinforcement of natural and artificially cemented sand, showing plastic wastes improved soil mechanical response. Later, several studies reported the influence of PET fiber inclusion on the mechanical properties of soils [17–21]. The behavior of soils reinforced with PP fibers has also been extensively studied [12,22–28]. However, there is a lack of studies involving the inclusion of polymeric strips taken from recyclable materials as soil reinforcement.

The use of polymeric strips has several advantages, such as the possibility of reusing plastic waste to increase soil strength without the need to apply a recycling process, as in the case of synthetic fibers. However, the few available studies used PET strips and not PP strips, e.g., [2,17,29–32].

Sivakumar Babu and Choukey [17] evaluated the effect of including 12 mm long and 4 mm wide PET strips, in amounts of 0.50%, 0.75%, and 1.0%, in a sandy soil using unconfined compression strength (UCS) tests and triaxial tests (consolidated and undrained). The authors reported significant increases in soil shear strength parameters, which were greater for larger numbers of strips. In addition, UCS tests indicated an increase in ductility, proportional to the inclusion of strips. Soltani-Jigheh [31] studied the inclusion of PET strips (4 mm wide and 8 mm long) in quantities of 0.25%; 0.50%; 0.75%; 1.0%; 1.5%, and 2% (in relation to the clay soil mass) using consolidated undrained (CU) triaxial tests. Results showed an increase of around 11% in the shear strength of the soil, resulting from an increase in apparent cohesion and a decrease in friction angle.

Babu and Choukey [17] suggested a more economic and simple way of recycling plastic bottles as soil reinforcement using strips cut from PET water bottles. Plastic strips that were 12 mm long and 4 mm in width showed significant improvement in the strength of two soils due to an increase in friction and significant reduction in compression parameters. Chebet and Kalumba [30] evaluated soil improvement using HDPE plastic strips (0.1–0.3% by weight, 15 to 45 mm length, and 6 to 18 mm widths) obtained from shopping bags mixed with two sandy soils through direct shear tests. Findings showed that shear strength of sandy soils was sensitive and significantly affected with a small addition of strips. Luwalaga [2] evaluated sand reinforced with randomly mixed PET plastic waste flakes with varying percentages in terms of California Bearing Ratio CBR and direct shear box testing. Results concluded that the appropriate percentage of PET plastic waste to use while reinforcing sandy soil used is 22.5%. Peddaiah et al. [32] evaluated the addition of PET wastewater bottles cut into strips to locally available soils and showed enhanced soil engineering properties. Strips were cut with 15 mm width and lengths of 15, 25, and 35 mm in different contents of 0.2% to 0.8%. Strips randomly mixed with sandy soil improved the soil strength parameters. It was found that addition of PET strips to sand could reduce the soil brittleness under low overburden pressures.

According to Fathi et al. [33] recycling plastic waste as reinforcing material has become a cheap and viable alternative for soil improvement. Peddaiah et al. [32] concludes that the effect of plastic reinforcement in soil mass vitally depends on nature of the surface (i.e., plain/smooth or corrugated/undulated) and size of strips, plastic content, and type of soil. For Onyelowe et al. [34] the fundamental purpose of solving an engineering problem revolves around a sustainable, economic,

efficient, and durable design, with optimal performance to meet certain desirable conditions. Hence, the sustainable and economic alternative of plastic waste strips and local soils offers two advantages in geotechnical applications: reuse of plastic waste materials and reduction in the use of natural soils, producing materials with required engineering properties.

Although the use of strips from the reuse of waste bottles has high potential for improving soil characteristics, the field of study for these materials is relatively new, especially regarding lateritic soils. This fact generates a consensus among several authors regarding the need for a deeper assessment of different types of plastics and the characteristics of each type of inclusion in conjunction with different soils, in addition to full-scale studies [2,30,32].

Considering the experience from the literature, as well as the lack of research regarding polymeric strips as soil reinforcements, the strength properties of compacted lateritic soils reinforced with polypropylene waste strips cut from recycled plastic packing were evaluated in this study. A series of unconfined compressive strength (UCS), direct shear, and California bearing ratio (CBR) tests were conducted in order to evaluate an optimum combination of plastic waste strips in different soils. A statistical analysis of proposed equations to estimate the UCS of PP strip-stabilized soils is presented. Results were used to prepare samples for CBR and direct shear tests.

## 2. Materials and Methods

Lateritic soils (clayey sand and clay) were chosen in this research, since they represent typical soils that cover a large area in Brazil. These soils are residual sandstone soils, with low compressibility, unsaturated condition, and high porosity. The clayey sand was collected in Bauru, Sao Paulo, Brazil (22°21′6.03″ S; 49°01′57.68″ O), and the clay soil was collected in Pederneiras, also in Sao Paulo state (22°19′52.5″ S; 48°45′32.26″ O). The soil samples were characterized according to the following American Society for Testing and Materials (ASTM) recommendations: particle size analysis ASTM D7928 [35], soil classification (USCS) ASTM D2487 [36], Highway Research Board (HRB) classification ASTM D3282 [37], specific gravity ($G_s$) ASTM D854 [38], Proctor tests ASTM D698 [39], and consistency limits ASTM D4318 [40]. The physical properties of the soils including their classification from these tests are presented in Table 1. The particle distributions and the standard Proctor compaction test results for the sandy clay (SC) and clay low (CL) soils are shown in Figures 1 and 2, respectively.

**Table 1.** Physical properties of soils used in this research.

| Property Value | Clayey Sand | Clay | Specification |
|---|---|---|---|
| Soil classification (USCS) | SC | CL | ASTM D2487 [36] |
| HRB classification | A-2-4 | A-6 | ASTM D3282 [37] |
| Percent sand (%) | 80 | 8 | ASTM D7928 [35] |
| Percent fines (<0.074 mm) (%) | 20 | 92 | |
| Specific gravity, Gs | 2.65 | 2.69 | ASTM D854 [38] |
| Maximum dry unit weight (kN/m$^3$) | 19.50 | 18.4 | ASTM D698 [39] |
| Optimum water content (%) | 10.6 | 16.1 | |
| Liquid limit | 16 | 34 | |
| Plasticity limit | NP | 23 | ASTM D4318 [40] |
| Plasticity index | NP | 11 | |

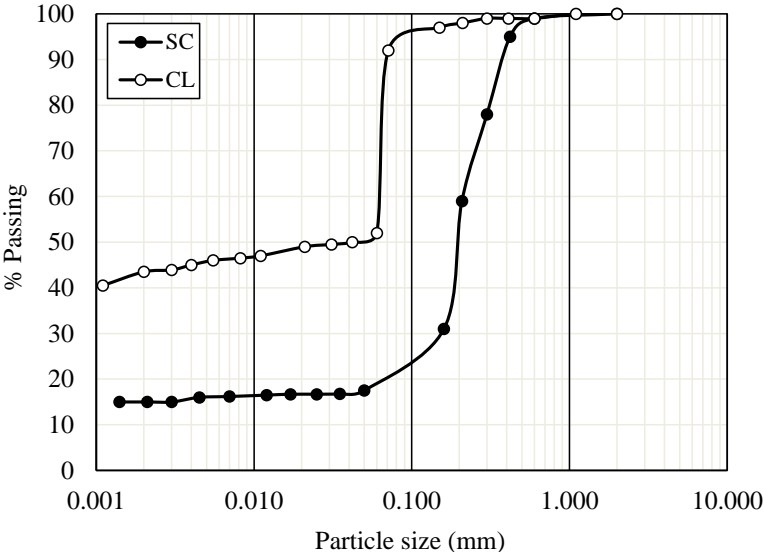

**Figure 1.** Particle size distribution of the two lateritic soils.

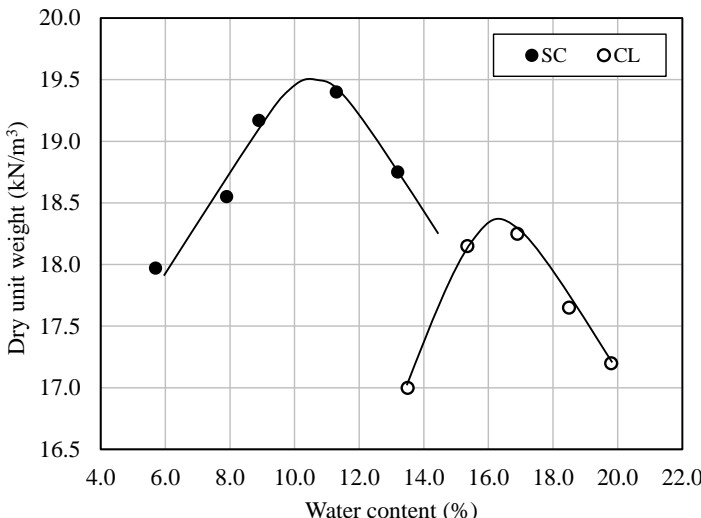

**Figure 2.** Compaction curves of the two lateritic soils under investigation.

The soil/water retention curves (SWRCs) of the two soils are presented in Figure 3, along with the fitted SWRC model of van Genuchten [41]. The SWRC data exhibit a bimodal behavior (two air entry suctions), while the van Genuchten [41] SWRC is unimodal, as follows:

$$w = w_r + (w_s - w_r) \times \left\{ \frac{1}{[1 + (\alpha.s)^n]^m} \right\} \tag{1}$$

where $w_s$ and $w_r$ are the saturation and residual water content (%), $m$ and $n$ are curvature parameters, and $s$ is the matric suction (kPa).

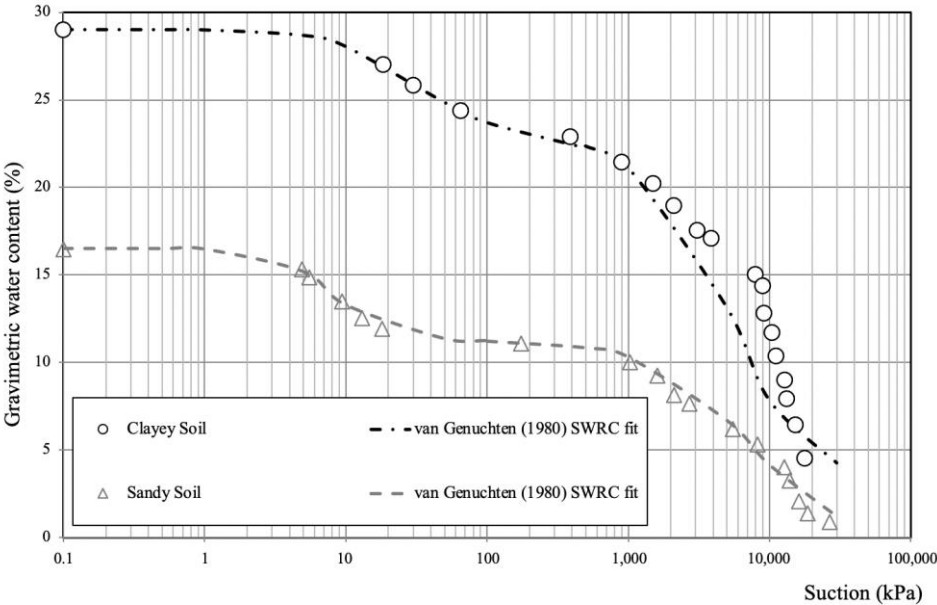

**Figure 3.** Soil water retention data for the two soils: sandy soil and clayey soil.

Accordingly, the van Genuchten [41] SWRC was fitted to both of the modes exhibited in the data. Specifically, the fits were performed in two parts for each curve. This behavior can be attributed to the presence of macro and micropores in the soil [42]. The fitting parameters of the SWRC of van Genuchten [41] are shown in Table 2. The curve for the clayey sand (SC) soil shows two air entry suctions, the first of approximately 3 kPa, and the second of approximately 2 MPa. The curves obtained for the clay (CL) soil, due to the greater retention capacity, show a great variation of suction pressures over a small range of gravimetric water content. Similar to the SC soil, two air entry suctions are observed for the CL soil, the first of approximately 11 kPa, and the second of approximately 6 MPa.

**Table 2.** Fitting parameters of the van Genuchten (1980) soil/water retention curve (SWRC).

| Soil | Stretch | $\alpha$ (kPa$^{-1}$) | $m$ | $n$ | $w_r$ (%) | $w_s$ (%) | R-Squared |
|------|---------|------------|-----|-----|-----------|-----------|-----------|
| Sandy | 1 | 0.1520 | 0.6977 | 2.4762 | 11.2 | 16.5 | 0.996 |
|       | 2 | 0.0001 | 1.4349 | 1.1890 | 0.0 | 11.3 | 0.976 |
| Clayey | 1 | 0.0669 | 0.3421 | 1.8113 | 21.4 | 29.0 | 0.985 |
|        | 2 | 0.0003 | 0.4974 | 2.4974 | 3.00 | 22.6 | 0.976 |

Polypropylene (PP) strips were obtained from plastic packaging that would be discarded without any reuse. In order to avoid discrepancies in the results, only one specific brand of plastic packaging was used (without lids, labels, and other parts) in order to assure strip homogeneity. PP strips of 1.5 mm width and 0.5 mm thickness with lengths of 10, 15, 20, and 30 mm were added to the soil in different percentages by dry soil weight of 0.25%, 0.5%, 0.75%, 1.0%, 1.5%, and 2.0%. In order to achieve a uniform mixture, the soil and strips were homogenously distributed and mixed with the soil by hand mixing dry soil, water, and strips. The cutting process of the PP strips, the final shape of the strips, and an example of soil mixed with strips are shown in Figure 4. To prevent floating of the strips, the water was added before the strips. In addition, the specimens were destroyed after testing to verify segregation. In this sense, the visual inspection showed that the process of mixing the strips and soil provided an excellent integration of soil and strips. In field applications, mixing is performed according to the recommendations of Falorca et al. [43] and Shukla [14]. The aspect ratios ($A_r$) for the strips with a length of 10, 15, 20, and 30 mm are 20, 30, 40, and 60, respectively. The PP strips have a specific mass of 0.91 g/cm$^3$, a tensile strength of 150 MPa, and a tensile modulus of 3.5 GPa.

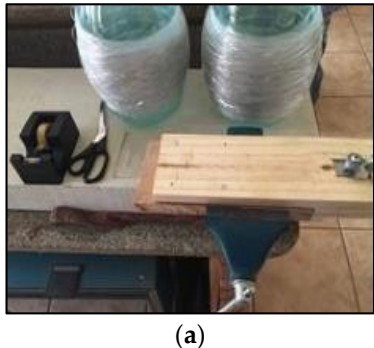 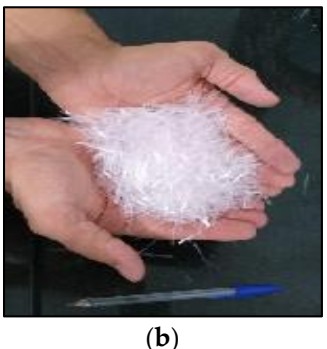 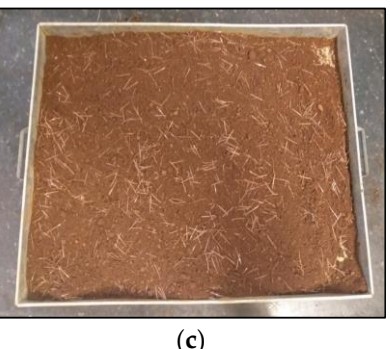

(**a**) (**b**) (**c**)

**Figure 4.** Polypropylene (PP) strips: (**a**) cutting process; (**b**) PP strips after cutting; (**c**) soil mixed with PP strips.

This study involved a combination of UCS, direct shear, and CBR tests to investigate the effect of strips on soil improvement. The UCS tests were conducted according to ASTM D2166 [44] with samples compacted at the optimum water content for each soil shown in Figure 2. Considering the importance of compaction parameters for each soil mixture in unconfined compression strength, standard Proctor compaction tests were conducted for each soil–strip mixture in order to compact soil specimens for UCS and shear strength tests. However, no significant alterations were observed in maximum dry unit weight and optimum water content (OWC) with PP strip addition, and the soil–strip samples were compacted at the OWC of natural soil conditions (Table 1). In order to examine the variability of the effect of waste strips in the UCS properties of both lateritic soils, triplicate specimens of 50 mm diameter and 100 mm height were tested. For each combination of optimum strip content obtained from the UCS results, drained direct shear tests were conducted according to ASTM D3080 [45] on the compacted unsaturated soils. Samples were consolidated under vertical stresses of 30, 60, and 125 kPa prior to shearing. Finally, CBR tests were conducted for each percentage of PP strips according to ASTM D1883 [46]. The specimens to be tested were also prepared with soil–strip samples compacted with optimum strip content properties in relation to UCS results.

## 3. Results and Discussion

### 3.1. Influence of PP Strips on Soil Unconfined Compression Strength (UCS)

The axial stress–strain curves from the UCS tests on the SC soil reinforced with PP strips are shown in Figure 5. Similar stress–strain curves were obtained for the CL soil. The curves in Figure 5 generally show that an increase in the peak value (the UCS) is observed after addition of PP strips. For both soils the results show that there is no big difference in the axial stress among the PP–soil mixtures as well as the pure soil before 2.5% of the axial strain. This behavior is in accordance with the literature [12,14,47]. As reported by Tang et al. [12], the addition of fibers does not affect the initial stiffness of unreinforced soil. Heineck et al. [47] concluded that the stiffness of soil–strip PP composite is not influenced at small strains. Shukla [14] states that only after a certain level of shear strain do the fibers begin to be more effective. The use of PP strips contributed to a change in the soil behavior from a brittle failure to a ductile failure, as shown in typical post-test photographs in Figure 6.

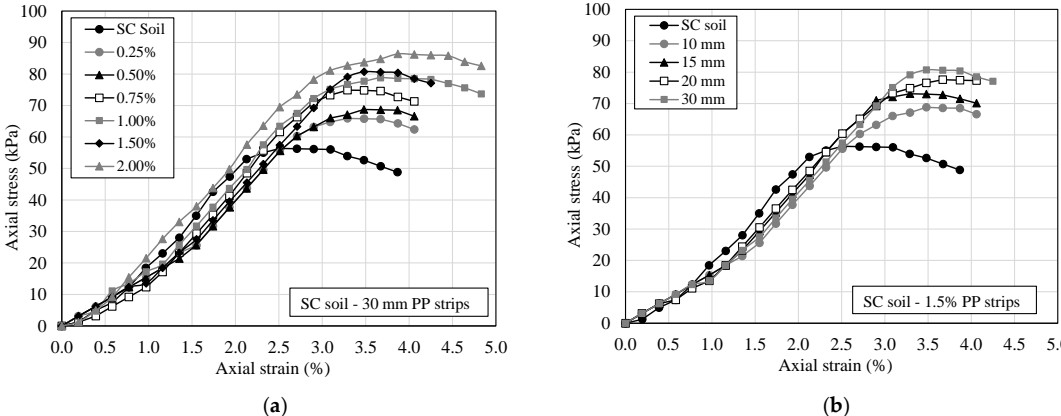

**Figure 5.** Axial stress–strain curves of clayey sand (SC) soil and PP strips: (**a**) increasing PP strip content; (**b**) increasing strip length.

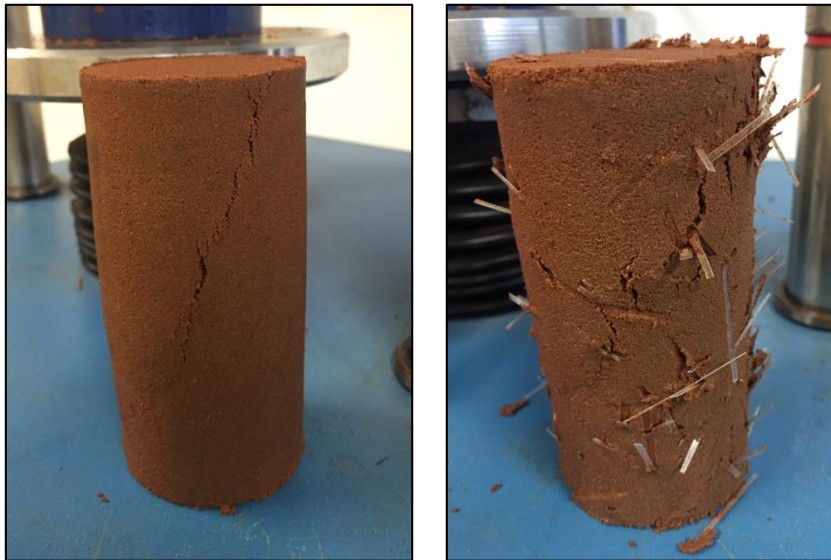

**Figure 6.** Specimens of natural and SC soil strips after failure.

The UCS values are shown in Figure 7 for the SC and CL soils as a function of PP strip contents for different strip lengths. For both soils, an increase in UCS was observed with increasing strip contents and lengths. No suction effects on strips results were noted. This can be explained by the fact that the strips are inert to the soil as well as by the gravimetric water content. An optimum combination of strip content and length was obtained for each soil from the UCS results. According to Figure 7a, the optimum combination for SC soil is 2% PP and 30 mm in length. In Figure 7b, the optimum combination for CL soil is 1.5% PP and 30 mm in length. These values were adopted, since previous UCS tests performed using contents of 2.5% and 3.0% of strips led the UCS values to a sharp drop for SC and CL soils considering all lengths and contents. Note that for CL soil (L = 20 and 30 mm) this occurred even before reaching 2.5%. This is probably due to the dimensions of the specimen, the length of the strips, and the excess of strips that accumulated in a concentrated manner in specific points in the specimen. Samples exhumed after the tests showed this agglomeration of the strips. This excess of strips complicates the process of compacting the specimens and can lead to lower density values, which decreases the UCS value. These results are in accordance with the literature; that is, the strength of fiber-reinforced soil increases with increasing aspect ratio of fibers [10].

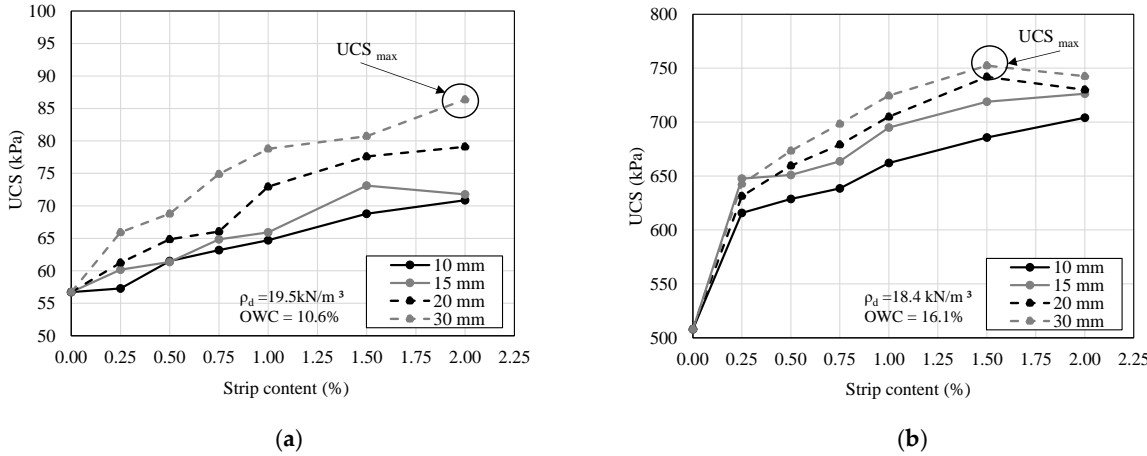

**Figure 7.** Unconfined compressive strength (UCS) results for different soils as a function of PP strip content for different PP strip lengths: (**a**) SC soil; (**b**) clay (CL) soil.

The UCS results for the two soils with different strip contents and strip lengths are shown in Figure 8. Both soils (with and without strips) were compacted at the corresponding optimum water content. It was observed that the soil highly influenced maximum UCS results. The SC soil presented higher increase in strength for increasing strip contents and length, showing that the soil friction is mobilized before mobilization of tension in the plastic strips. Higher strip lengths also indicated higher increase in SC shear strength, reaching the same strength increase of the clayey soil with 30 mm strip length. For the clayey soil, low contents of strips presented a significant strength increase, despite strip lengths. The increase in strip content also showed an increase in UCS.

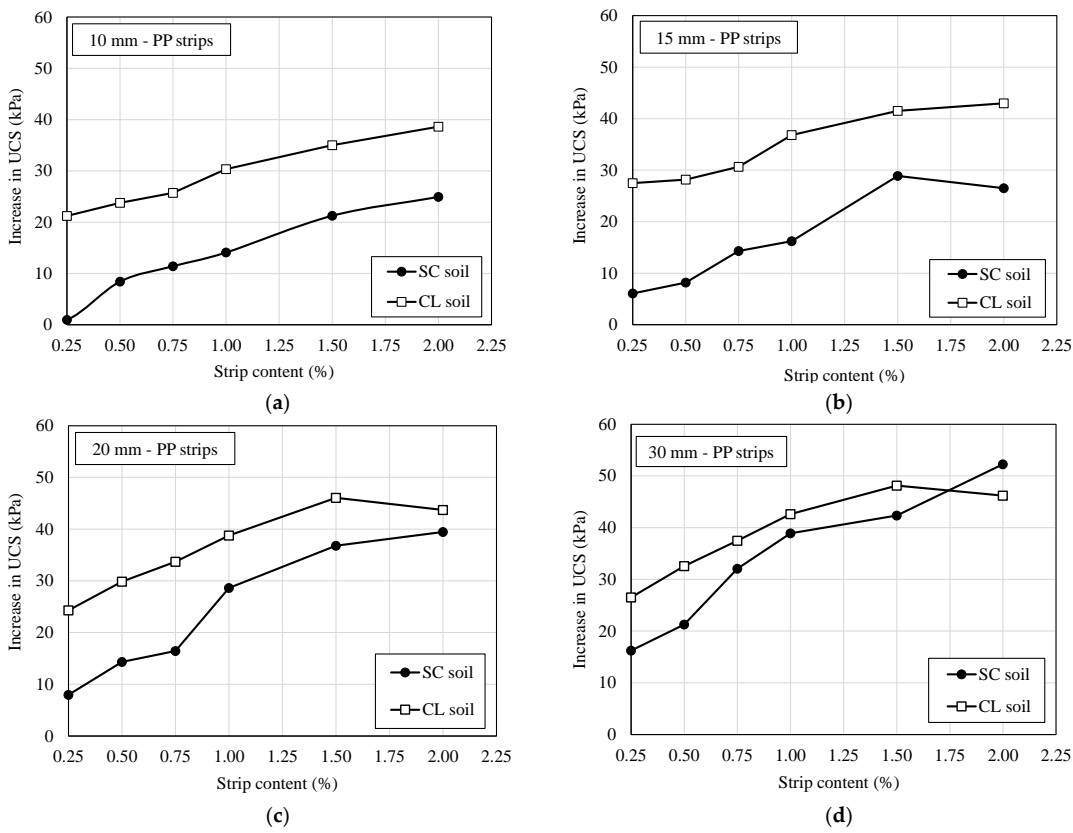

**Figure 8.** Influence of soil type on the UCS of soils with different strip lengths as a function of strip content: (**a**) 10 mm; (**b**) 15 mm; (**c**) 20 mm; (**d**) 30 mm.

As discussed, there are no results from the literature that discuss the use of PP strips in soil reinforcement. The literature only presents results of research using PP fibers. However, it can be seen that the results of this research are in accordance with previous results from the literature that evaluated PP fibers, e.g., [12,48–50]. Santoni et al. [48], for instance, concluded that the inclusion of randomly oriented discrete PP fibers significantly improves the UCS of sands. An optimum fiber length of 51 mm was identified for the reinforcement of sand specimens. A maximum performance is achieved at the fiber content between 0.6% and 1% by dry weight. The specimen performance is enhanced in both wet and dry optimum conditions. Tang et al. [12] evaluated the UCS of clayey soil cylindrical specimens (diameter = 39.1 mm, length = 80 mm) with inclusion of different contents of PP fibers (12 mm long). Fiber inclusion with 0.05% fiber content enhances the unconfined compressive/peak strength of soil. Kumar and Singh [49] used the random inclusion of PP fibers to evaluate the UCS of fly ash. At an aspect ratio (Ar) of 100, the unconfined compressive strength of fly ash increased from 128 to 259 kPa with an increment in fiber content from 0 to 0.5%. The results show that the variation of unconfined compressive strength with fiber content is linear, and the optimum fiber length and aspect ratio were found to be 30 mm and 100, respectively. Zaimoglu and Yetimoglu [50] investigated the UCS of fine-grained soil (MH, high plasticity soil) effects using randomly distributed PP fiber reinforcement (length = 12 mm; diameter = 0.05 mm). The main findings show that there is a tendency for UCS values to increase due to the increase in fiber content. The soil reinforced with a fiber content of 0.75% showed an expressive increase of 85% in the UCS value when compared to unreinforced soil. As Tang et al. [12] also discussed in their study, the increase in UCS might be due to the bridging effect of fiber, which can efficiently prevent the further development of failure planes and deformations of the soil.

The results from an analysis of variance (ANOVA) shown in Figure 9 indicate that the UCS is more affected by strip length or content. Results showed that strip content has a greater effect on results than strip length for both soils evaluated in this research. The equations were used to propose an analytical model to predict UCS of SC and CL soils reinforced with PP strips based on experimental results. The good agreement between the experimental data and the estimates indicates that the proposed model is adequate for estimating preliminary soil–strip UCS strength parameters. The limitations of the models include the type of soils used and PP strips with 15 mm width.

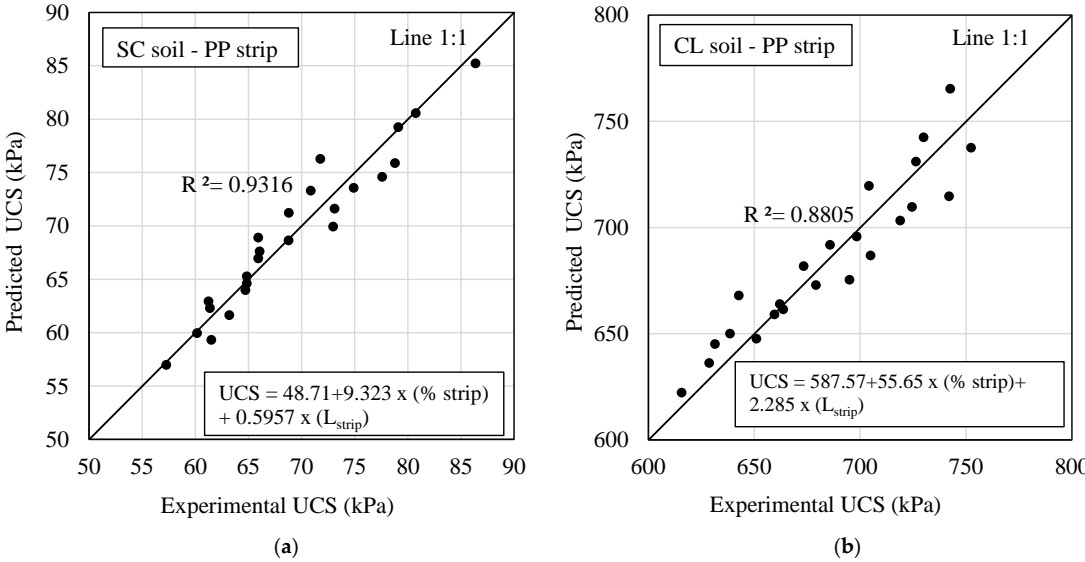

**Figure 9.** Prediction model for UCS of soil–strip mixtures: (**a**) SC; (**b**) CL.

An analysis showing the influence of compaction water content in UCS of soil–strip samples is shown in Figure 10. Samples at the optimum water content (OWC) had the best combination of strip length and content for each soil (Figure 7). UCS values were compared with the same mixtures compacted at OWC −2% and OWC +2% also using optimum strip combinations. The water content

at compaction influenced the UCS of both soils. OWC −2% presented higher influence on UCS of both soils but with opposite results. Sandy soil showed superior UCS when compacted at OWC −2%, while clayey soil showed a lower increase in UCS. The best result for clayey soil in terms of UCS increase was seen for soil–strip samples compacted at OWC +2%. Results are more attributed to soil type than strip content.

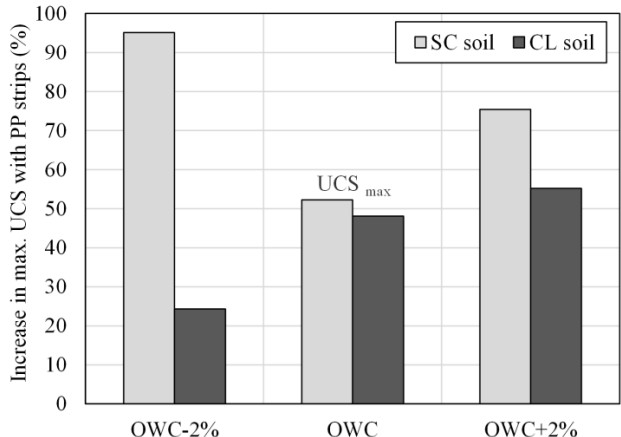

**Figure 10.** Influence of compaction water content on UCS results of soil mixtures at optimum strip combinations.

## 3.2. Influence of PP Strips on Drained Shear Strength

Results of direct shear tests considering each combination of soil and strips (15 × 30 mm) representing maximum UCS are presented in Figure 11. The specimens (with and without strips) were compacted at optimum water content. Figure 11a shows the shear strength envelopes of SC soil with and without PP strip reinforcement showing an increase in both apparent cohesion and friction angle. Figure 11b shows shear strength envelopes of the CL soil with and without PP strip reinforcement. In this case, results presented higher friction and no change in apparent cohesion. An improvement in shear strength parameters shown in Table 3 is observed with PP strip reinforcement, which can be attributed more attributed to friction than cohesion. Peddaiah et al. [32] showed results of increasing trend for apparent cohesion and friction angle with an increase in strip content, and the authors attribute this phenomenon to combined soil and plastic mass behavior during shearing. According to the author, increase in shear strength parameters is achieved because there is increase in frictional surface between soil particles and plastic strips.

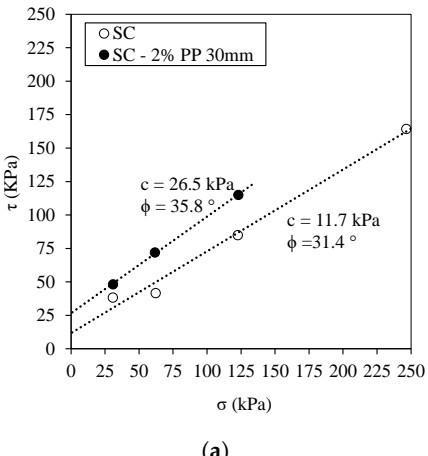

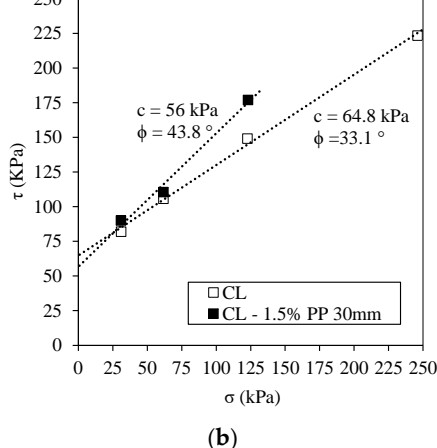

(a)                                   (b)

**Figure 11.** Shear strength envelopes of natural and PP strip-soils: (**a**) SC; (**b**) CL.

**Table 3.** Summary of shear strength parameters for polypropylene (PP) strips mixed with soils.

| Soil Type | PP Strip Content (%) | PP Strip Length (mm) | Effective Friction Angle (Degrees) | Increase in Effective Friction (%) | Apparent Cohesion (kPa) | Increase in Apparent Cohesion (%) |
|---|---|---|---|---|---|---|
| SC | 0.0 | 30 | 31.4 | NA | 11.7 | NA |
| SC | 2.0 | 30 | 35.8 | 1.18 | 26.5 | 2.26 |
| CL | 0.0 | 30 | 33.1 | NA | 56 | NA |
| CL | 1.5 | 30 | 43.8 | 1.47 | 64.8 | 0.86 |

It is important to note that, besides the fines contents, lateritic soils present good shear strength behavior when unsaturated. The natural clayey soil has a high friction angle (>30°), which is expected for lateritic soils. On the other hand, it is important to note that the soils are in an unsaturated condition, which could explain the high values of shear strength parameters, mainly the apparent cohesion (CL soil). The results presented in this research are in accordance with results of the literature, e.g., [12,14,43,51–54]. Falorca and Pinto [43] evaluated two soils very similar to the soils studied in this research. Authors carried out direct shear tests (60 mm square box) to evaluate the effect of short, randomly distributed PP microfibers on the shear strength behavior of two different types of soils: a poorly graded sandy (SP) soil and a clayey soil of low plasticity (CL). The main results show that the shear stress always increases up to the maximum deformation allowed, rather than reaching a peak or constant value typical for unreinforced soils. No significant difference was found when using straight or crimped fibers. The authors also concluded that the initial stiffness of the reinforced sand decreases with an increase in fiber content, whereas for reinforced clay there is no significant change. The reinforced sand is more compressive in the early stages of shear and more dilative subsequently compared with the unreinforced sand. There is much evidence that the influence of fiber content, fiber length, and normal stress level is due to the fibers' capacity to increase the number of contacts between soil particles and to mobilize a higher number of soil particles during shear. The number of fibers in the shear plane is a very important parameter.

Yetimoglu and Salbas [51] carried out a direct shear test (60 × 60 mm plane and 25 mm in depth) on sands reinforced with randomly distributed discrete PP fiber (length = 20 mm; diameter = 0.05 mm) reinforcements varying from 0.10% to 1%. The results of the tests indicated that the peak shear strength and initial stiffness of the clean, oven-dried, uniform river sand with particles of fine to medium size (0.075–2 mm) at a relative density of 70% were not affected significantly by the fiber reinforcement. Fiber reinforcements, however, could reduce soil brittleness providing smaller loss of post-peak strength and an increase in residual shear strength angle of the sand.

Tang et al. [12] conducted a series of direct shear tests on clayey soil cylindrical specimens (diameter = 61.8 mm, length = 20 mm) with inclusion of different percentages of PP fibers (12 mm long) at vertical normal stresses of 50, 100, 200, and 300 kPa. All the test specimens were compacted at their corresponding maximum dry unit weight and optimum water content. It was observed that the values of c and φ increased with increasing fiber content.

*3.3. Influence of PP Strips on Soil CBR*

Results of the CBR tests are shown in Figure 12. SC soil was highly influenced by plastic strips, with a 70% increase in CBR values. On the other hand, CL soil was not affected by strip inclusion, not altering CBR values.

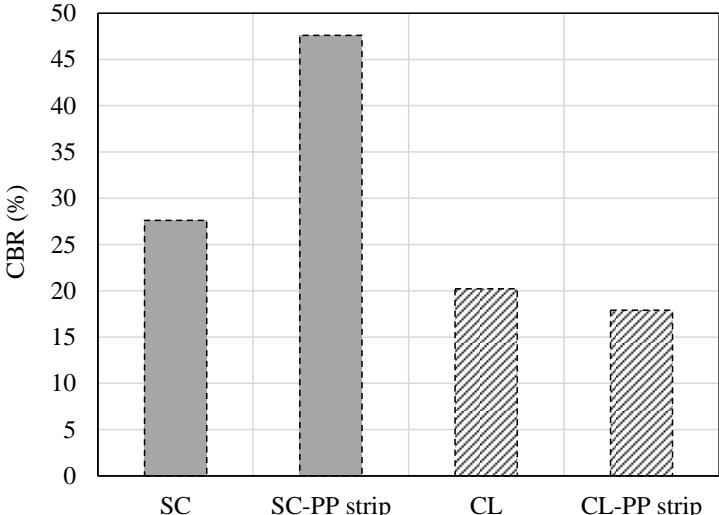

**Figure 12.** CBR values for SC and CL soils with and without PP strip reinforcement at their optimum combination identified from the UCS tests.

The results of the present research are in agreement with the results previously found in the literature for other soils and polymeric reinforcements, e.g., [49,50,55–58]. In this sense, as reported by Hoover et al. (1982), the CBR test values indicate that inclusion of fibers is most effective in sandy soils and less effective in fine-grained soils.

When evaluating the results obtained for the SC soil it was noted that they are in agreement with the results obtained by Fletcher and Humphries [55]. These authors showed that the CBR values of a silty soil increased significantly after the addition of PP fibers. According to the authors, PP fibers were used, and their content varied from 0%, 0.5%, 1%, and 1.5% in relation to the dry mass of soil, compacted with normal energy. The dimensions of the fibers used were 25 mm in length and 0.76 mm in diameter. According to the authors, there is an optimal fiber dosage that provides the highest CBR value. Higher than optimal dosages decrease the CBR value, since, with the increase in the number of fibers, there is a reduction in the amount of soil, which in turn affects the bonding forces at the soil–fiber interface. Finally, the authors concluded that the addition of fibers resulted in an increase in the CBR value of 133% when compared to the soil without the addition of fibers. Yetimoglu et al. [56] performed laboratory CBR tests to investigate the load-penetration behavior of a clean sand fill reinforced with randomly distributed discrete PP fibers (length = 20 mm; diameter = 0.50 mm) overlying a high plasticity inorganic clay with a nonwoven geotextile layer at the sand–clay interface as a separator. It was noticed that the peak load ratio (PLR) value increases with an increase in fiber content and becomes approximately five times as high as that of unreinforced sand.

Regarding the clayey soil, it was noted that the addition of fibers at the proposed optimum content generated an increase in expansion and a reduction in CBR due to the number of fibers present, impairing the contact (friction) between the particles. This behavior is in line with the results obtained by Pradhan et al. [58]. These authors evaluated the mechanical strength of a clayey soil reinforced with PP fibers by direct shear, unconfined compression, and CBR tests. The authors used PP fibers of 15, 20, and 25 mm in length with a 0.2 mm diameter, varying the fiber content from 0.1% to 1.0%, with an increase of 0.1%.

Chandra et al. [57] evaluated soils with PP fibers (length = 15, 25, 30 mm; diameter = 0.3 mm) and concluded that the CBR value of reinforced soils continues to increase with both fiber content and aspect ratio (Ar). However, they suggest that mixing soil and fibers is extremely difficult beyond the fiber content of 1.5%. The authors also suggest that 1.5% fiber content and an aspect ratio of 100 can be considered optimum values in the case of soils of low compressibility (classified as CL and ML), whereas 1.5% fiber content with an aspect ratio of 84 was found to be optimum for silty sand (classified as SM). Similarly, Kumar and Singh [49] studied fly ash (classified as silt

of low compressibility, ML) with randomly distributed PP fibers. The soaked and unsoaked CBR values presented increases with an increase in fiber content at a particular aspect ratio (60, 80, 100, or 120). Zaimoglu and Yetimoglu [50] also investigated the effects of randomly distributed PP fiber reinforcement (length = 12 mm; diameter = 0.05 mm) on the soaked CBR behavior of a fine-grained soil (MH, high plasticity soil) by conducting a series of CBR tests. The main results show that the CBR value presented a significant increase with increasing fiber content up to around 0.75% and remained more or less constant thereafter.

According to the design of flexible pavements [59], based on CBR values of pavement layers, a subgrade thickness for the SC soil used in this area of research (CBR = 28%) is 16 cm for heavy traffic condition (55 kN wheel load), and it reduces to 10 cm for the same traffic condition for 2.0% plastic waste mixed with soil (CBR = 48%). The final reduction implies the reduction of natural resources (aggregate materials) and construction costs. The clayey soil–strip mixture does not meet the required 20% CBR for subbases but can be suitable for other applications.

## 4. Conclusions

An extensive experimental program was conducted in order to assess the effect of polypropylene waste strips (cut from recycled plastic packing) mixed with lateritic soils. The experimental program involved the evaluation of soil UCS properties and an optimum combination of soil–PP strips. Outcomes of these combinations were used in CBR and shear strength analysis. The following conclusions can be drawn from this research:

- The use of PP strips as reinforcements in both SC and CL lateritic soils led to an increase in UCS, as well as a clear influence of PP strip length on the soil stiffness. The use of PP strips contributed to a change in soil failure from a brittle to a ductile mode;
- The UCS results revealed an optimum combination of PP strip content and strip length: SC soil and 2% PP and 30 mm in length and CL soil with 1.5% PP and 30 mm in length. The SC soil had a higher increase in UCS for increasing strip content and strip length, indicating that the soil friction is mobilized before strip mobilization. For the CL soil, low strip contents led to a significant increase in UCS regardless of the strip length. Statistical analysis conducted showed that strip content has a greater effect on the UCS than the strip length for both soils evaluated;
- The compaction water content had an important effect on the UCS of both soils, although opposite effects were observed in the UCS for both soils when increasing and decreasing the compaction water content by +2% and −2% from the optimal value;
- Results from direct shear tests indicate that PP strip–SC soil showed an increase in both apparent cohesion and friction angle, while PP strip–CL soil presented a higher friction angle and no change in apparent cohesion;
- California bearing ratio (CBR) tests indicate that SC soil was highly influenced by plastic strips and experienced a 70% increase in CBR after reinforcement. On the other hand, the CBR of the CL soil was not affected by the addition of plastic strips.

**Author Contributions:** Conceptualization, R.M., P.C.L., N.d.S.C. and J.S.M.; methodology, R.M., P.C.L., N.D.S.C., H.L.G., R.A.R. and J.S.M.; formal analysis, R.M., P.C.L., N.d.S.C. and J.S.M.; investigation, R.M., P.C.L., H.L.G., R.A.R.; resources, R.M., P.C.L., N.d.S.C. and J.S.M; writing—original draft preparation, R.M., P.C.L., N.d.S.C., H.L.G., R.A.R. and J.S.M.; writing—review and editing, R.M., P.C.L., N.d.S.C. and J.S.M.; visualization, R.M., P.C.L., N.d.S.C. and J.S.M.; supervision, P.C.L., N.d.S.C. and J.S.M.; project administration, P.C.L., N.D.S.C. and J.S.M. All authors have read and agreed to the published version of the manuscript.

**Funding:** This research received no external funding.

**Acknowledgments:** The author are very thankful to the Capes/Print program and PROPG/UNESP.

**Conflicts of Interest:** The authors declare no conflict of interest.

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
