# Peer review of "Reinforcing Effect of Polypropylene Waste Strips on Compacted Lateritic Soils"

_sustainability, doi:10.3390/su12229572_

Round 1

Reviewer 1 Report

Dear Authors

Your abstract is clear and comprehensive, structure of the text follows IMRaD (Introduction/Motivation, Materials/Methods, Results and Discussion/Results). Conclusions are well supported by testing results. List of references might be build on more recent articles but most of referred papers seem to be relevant.

My only doubt is based on concern about adequacy of laboratory testing with regard to possibility of field applications. From my experience, a uniform distribution of strips may be achieved in the lab but is almost impossible in building site conditions. That brings some threats about limitations of results applicability for the building industry.

I'd suggest to provide some comments concerning that issue.

I'd also appreciate some comments concerning the conformity of achieved results with your engineering intuition. Some of your results (like the last conclusion about CBR) is not surprising as frictional soils (with higher friction angle) cooperate well with strips and cohesive soils (clays) do not gain much from spread reinforcement. 

Reviewer 2 Report

This paper investigated the strength properties of compacted lateritic soils reinforced with 15 polypropylene (PP) waste strips. Comprehensive laboratory tests and analysis have been conducted by the authors. It is very interesting and very readable. The following comments are provided to the authors for further improvement.

Comments:

  1. Line 43, when do the literature review of the rubber-recycled materials mixtures, the following recent work should also be included:
  • Indraratna, B., Qi, Y. & Heitor, A. (2018), ‘Evaluating the Properties of Mixtures of Steel Furnace Slag, Coal Wash, and Rubber Crumbs Used as Subballast’, Journal of Materials in Civil Engineering, ASCE, 30(1), 04017251.
  • Qi, Y., Indraratna, B. and Coop, M.R. (2019). Predicted Behaviour of Saturated Granular Waste Blended with Rubber Crumbs. ASCE International J. of Geomechanics, ASCE 19(8): 04019079.
  • Qi, Y. and Indraratna, B. (2020), ‘Energy-based approach to assess the performance of a granular matrix consisting of recycled rubber, steel furnace slag and coal wash’, Journal of Materials in Civil Engineering, 32 (7), 04020169, ASCE DOI: 10.1061/(ASCE) 17 MT.1943-5533.0003239.
  • Qi, Y., Indraratna, B. and Vinod, J.S. (2018), ‘Behavior of Steel Furnace Slag, Coal Wash, and Rubber Crumb Mixtures, with Special Relevance to Stress-dilatancy Relation’, Journal of Materials in Civil Engineering, ASCE, 30(11): 04018276.
  1. Line 151-152: How to achieve a uniform mixture of PP and soils? How to prevent the segregation?
  2. Line 166-167: why choose 30, 60, and 125kpa as the confining pressures?
  3. In figure 5, seems there is no big difference in the axial stress among the PP-soil mixtures as well as the pure soil before 2.5% of the axial strain. Why is that?
  4. There is no text to describe the Figure 6.
  5. Line 186-187: it is not convincing to give the comments that for SC soil the optimum content of PP is 2% and the optimum length is 30mm. Because it can be concluded from Fig. 7a that the UCS is increasing with PP content and the PP length, so if keep increasing the PP contents and PP length UCS of the mixture may also increase. While for CL soils, it seems when the lengths of PP are 10 and 15mm the UCS results keep increasing with the PP content, so if keep increasing the pp content, UCS results of the mixture may exceed the value for the mixture with 1.5% pp with 30 mm, so it is not accurate to conclude that 1.5% of PP 30 mm length is the optimum combinations with CL soils.
  6. What’s the Coef of determination, R-squared values for the fitting equations shown in Figure 9?
